# Synergistic Antibacterial Activity with Conventional Antibiotics and Mechanism of Action of Shikonin against Methicillin-Resistant *Staphylococcus aureus*

**DOI:** 10.3390/ijms23147551

**Published:** 2022-07-07

**Authors:** Qian-Qian Li, Hee-Sung Chae, Ok-Hwa Kang, Dong-Yeul Kwon

**Affiliations:** 1Department of Oriental Pharmacy, College of Pharmacy and Wonkwang Oriental Medicines Research Institute, Wonkwang University, Iksan 54538, Jeonbuk, Korea; aliceqql@163.com; 2National Center for Natural Products Research, School of Pharmacy, The University of Mississippi, Oxford, MS 38677, USA; chaeheesung83@gmail.com

**Keywords:** MRSA, shikonin, antibacterial activity, synergistic effect, antibiofilm activity, PBP2a, virulence factor, α-hemolysin

## Abstract

Methicillin-resistant *Staphylococcus aureus* (MRSA) is a troublesome pathogen that poses a global threat to public health. Shikonin (SKN) isolated from *Lithospermum erythrorhizon* (*L. erythrorhizon*) possesses a variety of biological activities. This study aims to explore the effect of the combined application of SKN and traditional antibiotics on the vitality of MRSA and the inherent antibacterial mechanism of SKN. The synergies between SKN and antibiotics against MRSA and its clinical strain have been demonstrated by the checkerboard assay and the time-kill assay. The effect of SKN on disrupting the integrity and permeability of bacterial cell membranes was verified by a nucleotide and protein leakage assay and a bacteriolysis assay. As determined by crystal violet staining, SKN inhibited the biofilm formation of clinical MRSA strains. The results of Western blot and qRT-PCR showed that SKN could inhibit the expression of proteins and genes related to drug resistance and *S. aureus* exotoxins. SKN inhibited the ability of RAW264.7 cells to release the pro-inflammatory cytokines TNF-α and IL-6, as measured by ELISA. Our findings suggest that SKN has the potential to be developed as a promising alternative for the treatment of MRSA infections.

## 1. Introduction

*Staphylococcus aureus* (*S. aureus*) is a commensal pathogen in healthy adults and individuals with immunosuppressive or genetic predispositions [1]. *S. aureus* is a common cause of pneumonia worldwide, second only to pneumococcus [2]. *S. aureus* can be carried on the mucosa asymptomatically for weeks or months [3]. This is the leading cause predisposing colonized individuals to subsequent surgical site infections [4]. As a major human causative agent, *S. aureus* usually colonizes the anterior nares of humans and causes a wide range of intractable infections, ranging from mild superficial skin and wound infections to fatal disseminated infections [5,6]. The most common affected sites are skin and soft tissues. Infections at these sites include purulent infections such as furuncles, carbuncles, impetigo, folliculitis, and mastitis, as well as toxinogenic infections caused by staphylococcal scalded skin syndrome and Panton–Valentine leukocidin (PVL) [7].

Since the advent of penicillin in the 1940s, infectious diseases caused by *S. aureus* have been well controlled [8]. However, with the widespread use of penicillin, some *S. aureus* produce penicillinase, a specific type of β-lactamase, which can hydrolyze the β-lactam ring, making penicillin ineffective [9]. This prompted the development of a new semi-synthetic penicillin that is resistant to penicillinase, namely methicillin. The mechanism of action of methicillin differs from that of penicillin in that the additional methoxy group of methicillin produces an enzyme that can reduce the affinity for staphylococcal β-lactamase [10]. After methicillin was deployed to the clinic in 1959, it effectively controlled the infection of the penicillinase-producing strain of *S. aureus* [8]. Unfortunately, after two years, the first isolate of methicillin-resistant *S. aureus* (MRSA) was discovered in the United Kingdom [11]. 

In recent years, many clinically used chemotherapeutics have been derived from plant sources [12]. As a well-known herbal crop, *Lithospermum erythrorhizon* (*L. erythrorhizon*) is commonly used in ancient oriental remedies to treat burns, infectious crusts, bedsores, ulcers, sore throats, hemorrhoids, trauma, and exudative dermatitis due to its anti-inflammatory and antimicrobial activities [13,14]. Shikonin (SKN) (Figure 1) is a highly liposoluble naphthoquinone pigment extracted from the root of *L. erythrorhizon* [15]. SKN has been proven to have manifold biological activities. Recently, there have been many studies on the anticancer role of SKN. Zhang et al. revealed that SKN could prevent LPS-induced acute lung injury in mice by inhibiting myeloid differentiation protein 2 (MD2) [16]. Gara et al. clarified that SKN can be useful in the treatment of hormone refractory prostate cancer by regulating the pro-apoptotic endoplasmic reticulum stress and mitochondrial apoptosis pathways [17]. Chen et al. demonstrated that SKN inhibits proliferation and induces cell cycle arrest in human colorectal cancer by inhibiting hypoxia-inducible factor-1 α signaling [18]. The activity of SKN to suppress proliferation and promote apoptosis in various tumors has been repeatedly reported [19,20]. However, studies about the effects of SKN on MRSA infection have been insufficient. 

Our previous studies revealed the inhibitory effect of SKN on the vitality of MRSA strains and confirmed the synergistic effect of the addition of membrane permeabilizers or ATPase inhibitors on the efficacy of SKN [13]. Considering the excellent antibacterial effect of SKN, it is imperative to study more comprehensive molecular pharmacological mechanisms of SKN. In this study, we aimed to investigate the effects of the combined use of SKN and conventional antibiotics on the vitality of MRSA as well as the underlying antibacterial mechanisms of SKN, which might benefit future therapy for intractable MRSA infections.

## 2. Results

### 2.1. Synergistic Effects of SKN and Antibiotics against MRSA and Its Clinical Strain

This study determined the in vitro anti-MRSA activity of SKN in combination with conventional antibiotics. The results of the checkerboard assay showed that the combined use of SKN and seven conventional antibacterial agents, namely, gentamicin (GEN), amikacin (AMI), amoxicillin (AMO), cefoxitin (CEFO), ceftazidime (CEFT), and chloramphenicol (CHL), had a synergistic effect against MRSA and its clinical strain (Table 1). These data indicated that the MIC of SKN was reduced by 2-to 16-fold by combining with antibiotics against the standard MRSA ATCC 33591, CCARM 3090, and the clinical strain DPS-1. In turn, SKN synergized the antibacterial activity of antibiotics against these 3 types of MRSA. The MIC value of GEN, AMI, and AMO were all reduced by 4-fold when used in combination with SKN against ATCC 33591. When GEN and SKN were used in combination to treat CCARM 3090, SKN enhanced the anti-MRSA activity of GEN by markedly reducing the MIC value by 8-fold, showing excellent synergistic antibacterial ability. Moreover, by combining with SKN against the clinical strain DPS-1, the MIC of AMO was reduced by 2-fold, while the MIC of the remaining 6 antibiotics was reduced by 4-fold. In addition, although the combined use of SKN and OXA has no synergistic effect on the standard strain ATCC 33591, the results demonstrated a partial synergistic action between SKN and OXA for CCARM 3090 and the clinical strain DPS-1. In brief, checkerboard analysis revealed a general decrease in the MIC values for SKN and seven antibiotics, suggesting that there may be interactions between these substances that could restore MRSA susceptibility to drugs.

### 2.2. Time Kill Assay

Based on the results of the checkerboard assay, this study further verified the synergistic effect of SKN and antibiotics through time-killing analysis. According to the results depicted in Figure 2, during the first 8 h of culture, the vitality of MRSA and its clinical strains treated with SKN or antibiotics alone was significantly inhibited compared with untreated bacteria. In the subsequent culture process, the bacteria treated with the single compound gradually ignored the influence of the drug, and the number of bacteria began to increase at a trend close to that of the control group. The combined use of SKN (3/4 MIC) and antibiotic AMI (1/2MIC) reduced the concentration (CFU/mL) of standard MRSA ATCC 33591 to less than 3 log_10_ (Figure 2a), which can indicate that SKN and AMI have a synergistic anti-ATCC 33591 effect. The combination of SKN and GEN was observed as the most effective against the clinical strain DPS-1 (Figure 2c). After 24 h of incubation, the combined treatment of 1/2MIC SKN and GEN (1/2MIC) is sufficient to completely kill DPS-1. For CCARM 3090, higher concentrations of SKN (3/4MIC) and GEN (1/2MIC) are required in combination to achieve the purpose of sterilization (Figure 2b). As tested by the time kill assay, SKN showed a significant synergistic effect when combined with the three antibiotics selected in this experiment.

### 2.3. Effect of SKN on MRSA Cell Membrane Integrity and Permeability

In order to explore how SKN inhibits the proliferation of MRSA, we tested the leakage of MRSA nucleotides and proteins to verify whether SKN has an effect on the integrity of bacterial cell membranes. Since nucleotides and proteins are released after the plasma membrane rupture, these exudates were quantified by monitoring the absorbance at 260 nm and 280 nm. The results of the loss of 260 nm and 280 nm absorbing material in the cells of the clinical strain of MRSA DPS-1 treated with sub-inhibitory concentrations of SKN are shown in Figure 3a,b. The OD of the filtrates of bacterial cells exposed to the SKN revealed an increasing release of 260 nm and 280 nm absorbing material according to SKN concentration. The results of nucleotide and protein leakage confirmed the occurrence of membrane destabilization triggered by SKN. In addition, the membrane permeability was determined by the crystal violet uptake assay. The uptake of crystal violet was 29.3% in the absence of SKN, which increased by 91.5% at 7.8 µg/mL (1/2 MIC) of SKN treatment (Figure 3c). Although the treatments of 1/4MIC and 1/8MIC SKN made the crystal violet uptake much weaker than the treatment of 1/2MIC SKN, there was a slight increase compared with the control group.

### 2.4. Transmission Electron Microscopy (TEM)

TEM images of the MRSA DPS-1 confirmed changes in the cytoplasmic membrane and cell wall of the bacteria following exposure to SKN. As shown in Figure 4a, an intact septum was evident in the untreated MRSA strains. Similarly, no significant damage to the cell membranes of the GEN-treated bacteria was observed (Figure 4b). The cytoplasmic membrane of MRSA strains was significantly less intact than the control group after exposure to 1/2 MIC of SKN (Figure 4c). In addition, the structure of the cytoplasmic membrane and cell wall of the bacteria was found to be completely disrupted after the combined treatment of SKN with GEN at sub-inhibitory concentrations, and the cytoplasmic contents were evacuated (Figure 4d).

### 2.5. The Anti-Biofilm Formation Activity of SKN

The influence of SKN on *S. aureus* biofilm formation was evaluated at MIC and sub-MIC. As shown in Figure 5a, SKN effectively inhibited the biofilm formation of clinical MRSA DPS-1 in a dose-dependent manner. Intercellular adhesion A (IcaA) and fibronectin-binding protein A (FnbA) are important biofilm matrix components for the formation of *S. aureus* biofilms. Expression analysis of genes encoding surface-associated biofilm adhesins by qRT-PCR showed that SKN significantly downregulated the gene expression of surface adhesins such as *icaA* and *fnbA* compared with the untreated strains (Figure 5b).

### 2.6. Effect of SKN on PBP2a Expression

In this study, Western blot was used to analyze the effect of SKN on PBP2a, the main drug-resistant protein of MRSA. The expression profile of PBP2a after SKN treatment is shown in Figure 6a,b. GAPDH was used as an internal control (results not shown). The results indicated that the addition of SKN can significantly inhibit the expression of the PBP2a protein. In addition, through the qRT-PCR analysis of PBP2a related regulatory genes, we found that SKN also showed a significant inhibitory effect on the transcription level of mecA and mecR1 in a dose-dependent manner (Figure 6c).

### 2.7. Effect of SKN on Virulence Factor Expression

In order to validate the potential in vitro effects of SKN on the virulence factors of MRSA strains, Wwestern blot analysis and qRT-PCR were used to quantify the translation of the main virulence proteins and the transcription of their related regulatory genes in the clinical strain DPS-1. According to the immunoblotting bands depicted in Figure 7a,b, SKN markedly reduced the expression of α-hemolysin (HLA), staphylococcal enterotoxin A (SEA) and staphylococcal enterotoxin B (SEB) in a dose-dependent manner, compared with the untreated group. Additionally, as shown in Figure 7c, the results revealed that SKN significantly down-regulated the expression of genes responsible for the production of α-hemolysin (*hla*) and staphylococcal enterotoxin A (*sea*), compared to the untreated control. Further, after SKN treatment, the expressions of regulatory genes *agrA*, *RNAIII*, *saeS,* and *sigB* were down-regulated, respectively (Figure 7d–f).

### 2.8. Effects of SKN in the Pro-Inflammatory Cytokine Release of 264.7 RAW Macrophages Stimulated by MRSA

An ELISA was used to evaluate the effect of SKN on the secretion of tumor necrosis factor α (TNF-α) and interleukin-6 (IL-6) in 264.7 RAW macrophages stimulated by clinical MRSA DPS-1. As illustrated in Figure 8a,b, the secretion of TNF-α and IL-6 in macrophages was significantly increased after MRSA stimulation. However, the culture supernatants of DPS-1 grown under sub-inhibitory concentrations of SKN blocked the ability of *S. aureus* to induce these two cytokines in a dose-dependent manner.

### 2.9. Effect of SKN on the Hemolytic Activity of S. aureus Strains

The effect of sub-inhibitory concentrations of SKN on α-hemolysin activity was determined by a hemolysis assay. As illustrated in Figure 8c, compared with the normal blank group only treated with 1 × PBS, the treatment of MRSA culture supernatant triggered the rupture and lysis of rabbit red blood cells. However, compared with the culture supernatants without SKN, the addition of SKN with 1/8, 1/4, and 1/2 MIC induced a significant reduction in the activity of α-hemolysin dose-dependently, reaching 1.69-to 3.42-fold in clinical strain DPS-1.

## 3. Discussion

So far, MRSA has spread all over the world and has become one of the most significant pathogens among human infections, frequently occurring in both healthcare facilities and communities. Infections caused by MRSA are responsible for prolonged hospital stays, increased medical care costs, and high mortality rates [3]. The low-affinity penicillin binding protein 2a (PBP2a) is a unique, monofunctional DD-transpeptidase that catalyzes the cross-linking reaction of two adjacent glycan strands using peptide stems in peptidoglycan biosynthesis. PBP2a is the key resistance enzyme that enables MRSA to resist β-lactam antibiotics in the organism [21]. β-lactam antibiotics irreversibly acylate the active site serine, making penicillin binding proteins (PBPs) a lethal target. Because of its sheltered active site conformation, PBP2a is resistant to available β-lactam antibiotics, according to the X-ray structure. The activity of PBP2a is regulated by an allosteric site, distal (60 Å) from the active site [22]. The binding of cell-wall peptidoglycan to the allosteric site triggers the conformational opening of the active site [21,22]. Due to the absence of an allosteric trigger, it is difficult for β-lactam antibiotics to enter the active site, rendering them ineffective in infection treatment [23]. Essentially, PBP2a remains a competent catalyst for cell wall synthesis while resisting the modification of β-lactam antibiotics [24]. This study explored the expression of the PBP2a protein in MRSA treated with SKN, and its dose-dependent reduction proved the inhibitory effect of SKN on PBP2a, which is an important mechanism for SKN to reverse MRSA resistance. PBP2a is encoded by the *mecA* gene located on staphylococcal cassette chromosome *mec* (SCC*mec*). In qRT-PCR analysis, SKN suppressed the expression of *mecA* and its regulator, *mecR1*, and the higher the concentration of SKN, the better its inhibitory effect. In brief, SKN has shown the ability to inhibit the expression of PBP2a, an important resistance mechanism of MRSA, in both gene transcription and protein translation.

The use of a combination of phytochemical compounds and antibiotics may be a novel strategy to overcome the problem of antibiotic resistance. The results of the checkerboard test and time kill assay illustrated that the combination of SKN and conventional antibiotics was important to optimize the antimicrobial effect of both. This means that the antibacterial activity of each compound at low concentrations may be enhanced by another compound in the synergistic mixture. This combination has the potential to treat MRSA infection as a new antibacterial remedy. The strategy of using plant extracts in combination with traditional antibiotics has significant potential for the treatment of MRSA infection and reversion of MRSA resistance.

Antimicrobial compounds must penetrate or destroy the bacterial plasma membrane in order to kill bacteria. Any slight disturbance to the structural integrity and stability of the cell membrane could damage the normal metabolic function of the bacterial cell, thus affecting the cell vitality [25]. The action of antibacterial drugs, such as antibiotics, could cause a cytoplasmic burst in bacterial cells, resulting in the leakage of cellular contents. The capability of leaked nucleic acids (such as DNA and RNA) to absorb 260 nm wavelength ultraviolet light is here defined as 260 nm absorbing material, and 280 nm absorbing material is interpreted as leaking protein molecules capable of absorbing ultraviolet light at 280 nm wavelength [26]. In the present study, the release of 260 nm and 280 nm absorbing materials was used as a sensitive indicator of the cell membrane integrity. Compared with the control, the absorbance of the supernatant was significantly increased due to the release of nucleic acids (260 nm) and amino acids (280 nm) from the MRSA culture initiated by SKN. The results indicated that SKN might cause the loss of the integrity of the MRSA membrane, thereby increasing the permeability of cells to external protons and ions and ultimately leading to cell lysis and cell death. In addition, when the cell membrane is defective, it is easy to increase the uptake of crystal violet. Here, we observed that 1/2 × MIC SKN induced greater uptake of crystal violet dye compared with the untreated group, suggesting that the permeability of the membrane was altered by SKN and might lead to increased leakage levels. The measurement of 260 nm and 280 nm absorbing materials, coupled with the crystal violet uptake analysis, proved that SKN exhibits the ability to rupture cell membranes and increase permeability, which allows macromolecules to easily penetrate the membrane, leading to the leakage of intracellular components. Furthermore, the TEM image suggested that the combined treatment of SKN and GEN had a synergistic effect on the disruption of bacterial cell membranes.

The adhesion proteins expressed by *S. aureus* are important components of the attachment process and the development of the biofilm matrix. These adhesion proteins help bacteria adhere to the surface of host cells, thereby facilitating the formation of biofilms on implanted medical devices such as artificial joints, catheters, and heart valves. The formation of biofilms will increase the resistance of bacteria by hindering the penetration of antimicrobials, which makes the treatment of MRSA infections increasingly difficult [27]. Therefore, it is of great significance to study the formation of biofilm and find an anti-biofilm agent for MRSA that is conducive to the selection of the right treatment. In the present study, the crystal violet staining assay implied that SKN markedly inhibited biofilm formation dose-dependently, even at concentrations lower than MIC. The surface adhesins in the biofilm of *S. aureus*, such as intercellular adhesions (IcaA, D, B, C) and fibronectin-binding protein A (FnbA), are well characterized. The down-regulation of the expression of genes encoding IcaA (*icaA*) and FnbA (*fnbA*) indicated that SKN could inhibit the regulatory genes of biofilm formation. Therefore, we speculate that SKN has the ability to prevent the formation of MRSA biofilms.

This study also explored the effect of SKN at sub-inhibitory concentrations on the expression of bacterial virulence factors. The present study evaluated them through different methods since the changes in the expression of virulence factors induced by antibacterial agents are diverse. This study explored the effects of SKN at sub-inhibitory concentrations on protein translation and gene transcription of bacterial virulence factors. α-hemolysin (HLA) and staphylococcal enterotoxins (SEs) are the main virulence factors produced by *S. aureus* strains. HLA is a 33-kDa pore-forming toxin encoded by the *hla* gene that can cause *S. aureus* pneumonia, sepsis, skin necrosis, septic arthritis, hemolysis, brain abscess, and corneal infection [27]. Staphylococcal enterotoxin A (SEA) and staphylococcal enterotoxin B (SEB), the most common SEs, are encoded by the *sea* and *seb* genes, respectively. The expression of *S. aureus* virulence factors is strictly regulated by a network of regulatory systems [28]. In the present study, we analyzed two important regulatory systems, comprising the accessory gene regulators (*agr*) and staphylococcal accessory protein effectors (*sae*). The *agr* operon is considered to be the quorum sensing regulation system of *S. aureus*, which can up-regulate the expression of exotoxins such as HLA and SEs [27]. As the major effector of the *agr* system, RNAIII positively regulates the expression of virulence factors [29]. The *sae* locus is composed of a two-component signal transduction system encoded by *saeS* and *saeR*, which can up-regulate the expression of virulence factor genes at the transcription level [28]. Moreover, the production of *S. aureus* virulence factors is also controlled by the sigma factor σB encoded by the *sigB* gene. Our results from Western blot analysis demonstrated that the protein expression of HLA, SEA, and SEB was significantly inhibited by treating them with sub-inhibitory concentrations of SKN. According to the results of qRT-PCR, SKN at sub-inhibitory concentrations has been indicated to interfere with the expression of virulence factor regulatory genes, such as *agrA*, *RNAIII*, *saeS*, and *sigB*, which then affect the transcription of exoprotein-coding genes (such as *hla* and *sea*). In addition, SEs can act as superantigens to induce macrophages to release pro-inflammatory cytokines. The effect of SKN on the production of pro-inflammatory cytokines (TNF-α and IL-6) was assessed using an ELISA. As a result, SKN at sub-inhibitory concentrations significantly reduced the excessive release of TNF-α and IL-6 induced by *S. aureus* culture supernatants, elucidating the biological relevance of SKN-induced reduction of *S. aureus* virulence factors. Additionally, we investigated the influence of sub-inhibitory concentrations of SKN on the hemolytic ability of clinical MRSA strains. *S. aureus* can produce hemolysin to lyse red blood cells. The hemolysin assay indicated that the presence of MRSA culture supernatant triggered the rupture and lysis of rabbit erythrocytes. However, the hemolytic activity of MRSA was significantly reduced by SKN in a dose-dependent manner. 

In summary, SKN showed antimicrobial activity against MRSA and its clinical isolate (Figure 9). We found a significant synergistic effect of SKN and antibiotics from the combination of herbal medicine and orthodox medicine, which related to the ability of SKN to disrupt the cell wall of MRSA. The anti-MRSA synergy effects observed in this study may be linked to the inhibition of PBP2a expression and the effect on membrane function. Additionally, SKN inhibits the hemolytic activity of MRSA. Therefore, SKN could provide another alternative treatment for solving the problem of MRSA.

## 4. Materials and Methods

### 4.1. Bacterial Strains and Reagents

ATCC 33591 standard MRSA strain was purchased from the American Type Culture Collection (Manassas, VA, USA). CCARM 3090 was provided by the Culture Collection of Antimicrobial Resistant Microbes (National Research Resource Bank, Seoul, Korea). The clinical MRSA isolate (DPS-1) was collected from the patient at the Department of Plastic Surgery, Wonkwang University Hospital (Iksan, Korea). ATCC 33591 and DPS-1 were incubated on Mueller–Hinton agar (MHA) and suspended in Mueller–Hinton broth (MHB), while CCARM 3090 was incubated on Brain Heart Infusion agar (BHIA) and suspended in Brain Heart Infusion broth (BHIB). MHA, MHB, BHIA and BHIB were obtained from Becton, Dickinson and Company (Sparks, MD, USA). Shikonin (SKN), oxacillin (OXA), gentamicin (GEN), amikacin (AMI), amoxicillin (AMO), cefoxitin (CEFO), ceftazidime (CEFT), chloramphenicol (CHL), Thiazolyl Blue Tetrazolium Bromide (MTT), and crystal violet solution were purchased from Sigma-Aldrich Co. (St. Louis, MO, USA). Mouse anti-PBP2a antibody was purchased from DiNonA Inc. (Seoul, Korea). Rabbit anti-α-hemolysin, anti-Staphylococcus Enterotoxin A and anti-Staphylococcus Enterotoxin B antibodies were obtained from Abcam (UK). Mouse anti-GAPDH antibody was purchased from Santa Cruz (Dallas, TX, USA).

### 4.2. Checkerboard Dilution Assay

A synergistic interaction between SKN and antibiotics against MRSA was investigated using the checkerboard method [30]. The MRSA cultures (ATCC33591, CCARM 3090 and DPS-1) were grown in the presence of SKN at concentrations of 1/32 × MIC, 1/16 × MIC, 1/8 × MIC, 1/4 × MIC and 1/2 × MIC in combination with OXA, GEN, AMI, AMO, CEFO, CEFT, and CHL, with each concentration range from 1/32 × MIC to 1/2 × MIC. The fractional inhibitory concentration index (FICI) was calculated to quantify the synergy between SKN and antibiotics using the following formula:
$${FIC}_{antibiotic}=\frac{\mathrm{MIC}\mathrm{of}\mathrm{antibiotic}\mathrm{in}\mathrm{combination}}{\mathrm{MIC}\mathrm{of}\mathrm{antibiotic}\mathrm{alone}},{FIC}_{SKN}=\frac{\;MIC\mathrm{of}\mathrm{SKN}\mathrm{in}\mathrm{combination}}{\mathrm{MIC}\mathrm{of}\mathrm{SKN}\mathrm{alone}},FICI={FIC\;}_{\mathrm{antibiotic}}+{FIC}_{SKN}$$

FICI values were interpreted as follows: <0.5, synergy; 0.5–0.75, partial synergy; 0.75–1, additive effect; 1–4, no effect; >4, antagonism.

### 4.3. Time–Kill Assay

A time-kill assay was performed to confirm the synergistic antibacterial effect between SKN and antibiotics (GEN and AMI) [31]. The assay was performed at five different time intervals (0, 4, 8, 16, and 24 h). The bacterial suspensions (ATCC 33591, CCARM 3090, DPS-1) were added to MHB or BHIB containing either a single antibacterial compound (SKN or antibiotic) or a mixture of SKN and antibiotic (at synergistic antibacterial concentrations) to reach the final cell concentration of 1.5 × 10^6^ colony-forming units (CFU)/mL. The untreated strains were used as negative controls. After incubating for the corresponding time at 37 °C the surviving bacteria were appropriately diluted and inoculated on a drug-free MHA or BHIA plate to count the number of viable colonies. The growth curves were drawn based on the number of viable bacteria calculated by the dilution ratio. When a certain antibacterial compound reduces the bacterial concentration (CFU/mL) below 3 log_10_, it can be defined as a bactericidal agent [32].

### 4.4. Nucleotide and Protein Leakage

The effect of SKN on the cell membrane integrity of MRSA DPS-1 was measured following the method described by Ahmad et al. with some modification [26]. Overnight bacterial cultures (5 × 10^7^ CFU/mL) at 37 °C were washed and resuspended in 1 × PBS. Strains were treated with SKN (1/2 × MIC, 1/4 × MIC, 1/8 × MIC) while the untreated strains served as negative controls. After incubation at 37 °C for 2 h, cell suspensions were centrifuged at 13,000 rpm for 10 min, and the supernatants were collected and diluted appropriately. The optical density (OD) values at 260 nm (OD_260nm_) and 280 nm (OD_280nm_) were recorded using a spectrophotometer (Bio Tek Instruments Inc., Winooski, VT, USA).

### 4.5. Bacteriolysis Assay

The effect of SKN on MRSA cell membrane permeability was studied by measuring the uptake of crystal violet according to the modification of Meah et al. [32]. MRSA DPS-1 at 5 × 10^7^ CFU/mL was centrifuged at 4500 rpm for 5 min at 4 °C to collect the cell pellet. The pellet was washed three times and re-suspended in 1 × PBS to prepare a bacterial suspension. Sub-inhibitory concentrations (1/2 × MIC, 1/4 × MIC, 1/8 × MIC) of SKN were added to the cell suspension and incubated at 37 °C for 1 h. The bacterial suspension without SKN acted as a negative control. After 1 h of incubation, the suspensions were centrifuged at 13,000 rpm for 5 min at 4 °C to harvest the pellet. Then, the pellet was suspended in a crystal violet solution (10 µg/mL in 1 × PBS) and incubated at 37 °C for 10 min. After that, the suspensions were centrifuged at 13,000 rpm for 15 min to obtain the supernatant. The OD_590nm_ of the supernatants was recorded using a spectrophotometer (Bio Tek Instruments Inc., USA). Assume that the OD value of the originally used crystal violet solution is 100% uptake [33]. The percentage of crystal violate uptake was calculated by the following formula:


$$\frac{{OD}_{590nm}(\mathrm{treatment})}{{OD}_{590nm}(10\mu g/mL\;\mathrm{crystal}\mathrm{violet}\mathrm{solution})}\times 100$$


### 4.6. Transmission Electron Microscopy (TEM)

Transmission electron microscopy (TEM) was performed using MRSA DPS-1 [34]. The overnight bacterial cultures were diluted into MHB and continued to grow at 37 °C until they reached the mid-logarithmic phase of growth. The MRSA was treated with SKN or a mixture of SKN and GEN for 30 min. Cell pellets were collected by centrifugation at 13,000 rpm for 10 min. After removing the supernatant, the pellets were fixed by immersion in the modified Karnovsky fixative solution (2% paraformaldehyde and 2% glutaraldehyde in 0.05 M sodium cacodylate buffer, pH 7.2) overnight at 4 °C. After being post-fixed in 1% osmium tetroxide for 1.5 h at 4 °C, the specimens were subjected to an increasing series of ethanol washes for dehydration purposes. Then, the dehydrated samples were infiltrated with various ratios of propylene oxide and EMbed 812 resin (1:1 for 2 h, 1:3 overnight, and 0:1 for 2 h). The capsulated specimens were polymerized for 48 h at 60 °C. Polymerized specimen blocks were cut into 1-μm-thick sections using an ultramicrotome and stained with 2% uranyl acetate and Reynold’s lead citrate. The specimens were viewed with an energy-filtering transmission electron microscope (LIBRA 120; Carl Zeiss, Oberkochen, Germany) at 120 kV. The electronic signals transmitted were recorded using a 4k × 4k slow-scan charge-coupled device camera (Ultrascan 4000 SP; Gantan, Pleasanton, CA), which was attached to the electron microscope.

### 4.7. Anti-Biofilm Formation Assay

The inhibition of SKN on the formation of MRSA biofilms was determined by a crystal violet staining assay [35]. Briefly, MIC and sub-inhibitory concentrations (1/2 × MIC, 1/4 × MIC, 1/8 × MIC, and 1/16 × MIC) of SKN were prepared in MHB using a 96-well microplate. The inoculum of DPS-1 was adjusted to 1.5 × 10^6^ CFU/mL. The untreated cell suspension was used as a control. After incubating for 24 h at 37 °C, the suspensions were removed, and the wells were rinsed with 200 μL PBS to remove free-floating bacteria. After air drying, the biofilms formed by adherent cells at the bottom of the wells were stained with 0.1% crystal violet (200 μL). After incubating at room temperature for 30 min, the wells were thoroughly washed with PBS. Then the plate was de-stained with 96% ethanol (200 μL) and incubated for 15 min. The absorbance of the solution was read spectrophotometrically at OD_590nm_.

### 4.8. qRT-PCR

To analyze the mRNA transcription levels, qRT-PCR was performed as previously described [36]. MRSA DPS-1 suspensions (OD_600nm_ value of 0.7) were incubated with sub-inhibitory concentrations (1/2 × MIC, 1/4 × MIC, 1/8 × MIC and 1/16 × MIC) of SKN. After shaking the culture, the bacterial cultures were centrifuged at 13,000 rpm for 10 min to pellet the bacterial cells. Total RNA was extracted using the E.Z.N.A.^®^ bacterial RNA kit (OMEGA Bio-Tek, GA, USA) according to the manufacturer’s protocol. Equal mRNA amounts (1 μg) were determined by measuring the absorbance ratio at 260 nm and 280 nm using a NanoDrop spectrophotometer (Bio-Tek, Winooski, VT, USA). Then the mRNA was reverse-transcribed into the complementary DNA (cDNA) using the QuantiTect reverse transcription kit (Qiagen, Hilden, Germany) according to the manufacturer’s protocol. 20 μL reaction mixtures containing SYBR master mix (Applied Biosystems, Massachusetts, USA), primers, sample cDNA, and deionized water were set up to run PCR using the StepOnePlus real-time PCR system (Applied Biosystems, France). The primer sequences used to synthesize the DNA template are listed in Table 2. The qRT-PCR results were normalized to *16S*, the housekeeping gene for *S. aureus*.

### 4.9. Western Blot Assay

To analyze the protein translation levels, the Western blot assay was performed as previously described [37]. DPS-1 was grown at an OD_600nm_ of 0.6 in MHB and treated with graded sub-inhibitory concentrations (1/2 × MIC, 1/4 × MIC, 1/8 × MIC, and 1/16 × MIC) of SKN. After shaking the culture, the bacterial cells were collected by centrifuging at 13,000 rpm for 10 min and lysed with SMART™ bacterial protein extraction solution (Intron Biotechnology, Seongnam, Korea). The suspension was centrifuged at 13,000 rpm for 10 min to obtain the soluble protein. The amount of protein was quantified using the Bio-Rad protein assay reagent (Bio-Rad Laboratories, CA, USA). An equal number of sample proteins were loaded for 10–12% sodium dodecyl sulfate-polyacrylamide gel electrophoresis (SDS-PAGE) and transferred onto polyvinylidene difluoride (PVDF) blotting membranes. After blotting, the membranes were blocked with 5% skim milk for 2 h and subsequently probed with the primary antibody overnight at 4 °C. The loading differences were normalized with a mouse monoclonal anti-GAPDH antibody. After incubating with the secondary antibody for 1 h, TOPview™ ECL substrate (Enzynomics, Daejeon, Korea) was supplemented to the membranes, and the immunoreactive protein bands were visualized using the ImageQuant LAS-4000 mini chemical luminescent imager (GE Healthcare Life, Seoul, Korea).

### 4.10. Enzyme-Linked Immunosorbent Assay (ELISA)

ELISA was performed using previously described methods [29]. MRSA DPS-1 strains were cultured in MHB with sub-inhibitory concentrations (1/2 × MIC, 1/4 × MIC, 1/8 × MIC, and 1/16 × MIC) of SKN. The strains without SKN treatment served as negative controls. After 4 h of incubation, the supernatants were collected and filtered through 0.2 μm microfilters. RAW 264.7 cells were suspended in RPMI 1640 (supplemented with 10% FBS, 100 IU/mL penicillin and streptomycin) at a density of 10^6^/mL, and then evenly distributed to a 96-well cell culture plate (100 μL per well). When the cells adhered to the bottom of the plate, we removed the cell culture medium, and then added fresh RPMI 1640 medium (150 μL) and filtered bacterial supernatant (50 μL) to each well. After incubation for 16 h, the culture medium was collected and centrifuged at 1000 rpm for 5 min. The cytokines in the supernatant samples were detected using the Mouse TNF (Mono/Mono) and Mouse IL-6 ELISA sets (BD Biosciences, San Diego, CA, USA) according to the manufacturer’s instructions.

### 4.11. Hemolysis Assay

The measurement of hemolysis of rabbit red blood cells was performed by following the method of Duan et al. [38] with slight modifications. 2% rabbit erythrocytes were prepared by washing defibrinated rabbit blood (KisanBio, Seoul, Korea) with 1 × PBS. The DPS-1 strain was grown at an OD_600nm_ of 0.6 in MHB and treated with sub-inhibitory concentrations of SKN. 1 × PBS was used as a negative control, while Triton X-100 was used as a positive control due to its ability to cause complete hemolysis. After shaking the culture, the bacterial suspensions were centrifuged at 4500 rpm for 1 min. The supernatants were filtered through 0.22 μm microfilters. An equal volume of 2% rabbit erythrocytes and bacterial supernatant were mixed and incubated at 37 °C for 1 h. Then the mixture was centrifuged at 4500 rpm for 1 min to obtain the supernatant. The absorbance value of the supernatants was measured at OD_405nm_.

### 4.12. Statistical Analysis

All experiments were performed in triplicate. The results were expressed as the mean ± standard deviation (SD). The statistical software (IBM SPSS version 24, IBM Company, Chicago, IL, USA) was used to analyze the statistical differences and multiple mean comparisons between groups using one-way analysis of variance (ANOVA) and Duncan’s multiple range test (DMRT). Different letters indicate that there is a statistical difference between different groups (*p* < 0.05), while the same letter indicates that the difference is not significant (*p* > 0.05).

## 5. Conclusions

Overall, this study confirmed the synergistic effect of the natural compound SKN and seven common antibiotics, as well as investigating the possible mechanism by which SKN inhibits MRSA vitality. Based on the results, we speculate that SKN has the ability to reverse the drug resistance of MRSA and its clinical strains by disrupting the integrity of bacterial cell membranes, interfering with the formation of *S. aureus* biofilms, and inhibiting the expression of PBP2a and *S. aureus* virulence factors. We believe that SKN, as a novel phytochemical compound, has the potential to reduce the pathogenicity of MRSA. The findings on the internal mechanisms of SKN against MRSA suggest that SKN treatment may be important for the control and prevention of MRSA infection and colonization.

## Figures and Tables

**Figure 1 ijms-23-07551-f001:**
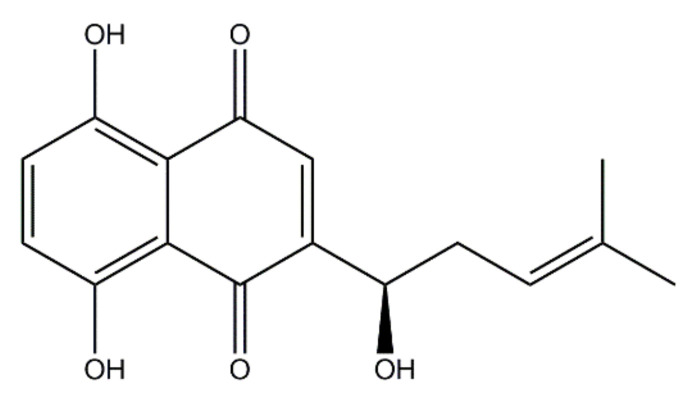
The chemical structure of shikonin (SKN).

**Figure 2 ijms-23-07551-f002:**
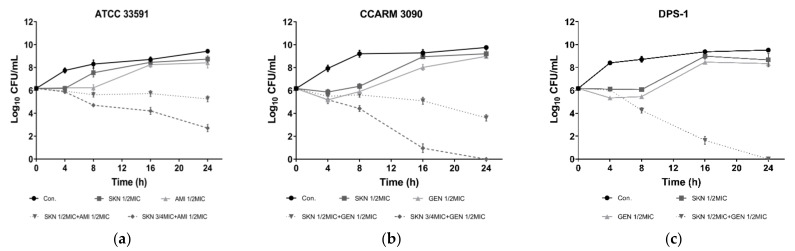
Time-kill curves of SKN and antibiotics alone and in combination against ATCC 33591 (**a**), CCARM 3090 (**b**) and DPS-1 (**c**). The data are means ± SD of triplicate determinations.

**Figure 3 ijms-23-07551-f003:**
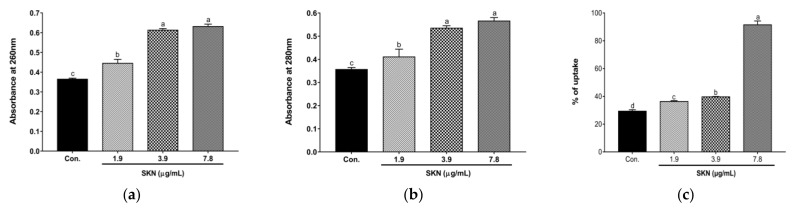
Cell contents release determined by measuring absorbance at 260 nm (**a**) and 280 nm (**b**) in DPS-1 strains treated with SKN. (**c**) Bacteriolytic effects of SKN against DPS-1. The data are means ± SD of triplicate determinations. Different letters on the bars indicate significant statistical differences between treatments (*p* < 0.05), while the same letters mean no significant differences (*p* > 0.05).

**Figure 4 ijms-23-07551-f004:**
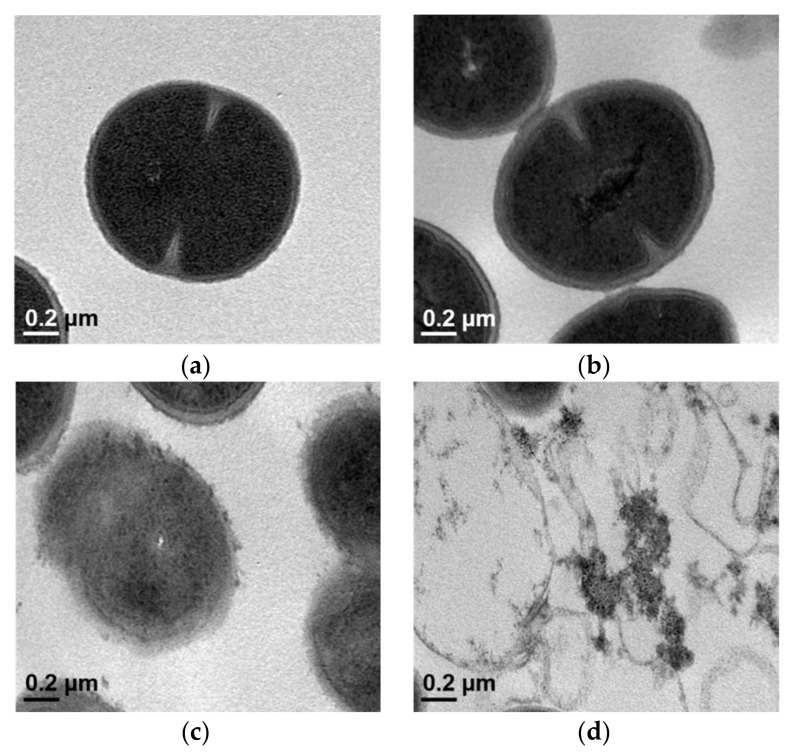
Transmission electron microscopy (TEM) analyses of MRSA DPS-1 after exposure to SKN. (**a**) Untreated control MRSA; (**b**) MRSA treated with 62.5 µg/mL GEN (1/2 × MIC); (**c**) MRSA treated with 7.8 µg/mL SKN (1/2 × MIC); (**d**) MRSA treated with 7.8 µg/mL SKN (1/2 × MIC) and 62.5 µg/mL GEN (1/2 × MIC).

**Figure 5 ijms-23-07551-f005:**
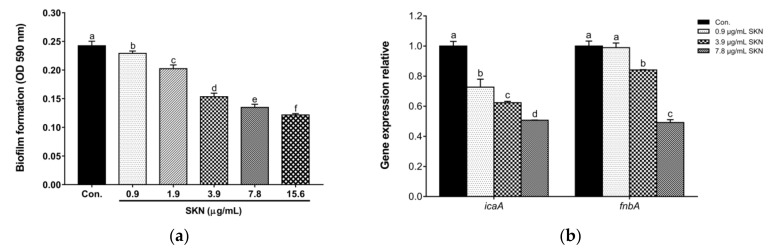
(**a**) The effect of SKN on the biofilm formation ability of MRSA DPS-1. (**b**) The effect of SKN on the expression of genes encoding surface-associated adhesins (*icaA* and *fnbA*). The data are means ± SD of triplicate determinations. Different letters on the bars indicate significant statistical differences between treatments (*p* < 0.05), while the same letters mean no significant differences (*p* > 0.05).

**Figure 6 ijms-23-07551-f006:**
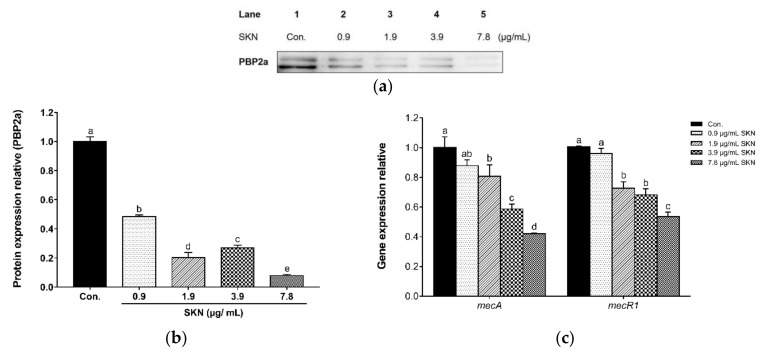
(**a**) The effect of SKN on the protein expression of PBP2a in clinical MRSA strain DPS-1, as analyzed by Western blot. (**b**) The intensities of Western blot protein bands were quantified using ImageJ software. (**c**) The effect of SKN on the mRNA expression of mec operon, as analyzed by qRT-PCR. The data are means ± SD of triplicate determinations. Different letters on the bars indicate significant statistical differences between treatments (*p* < 0.05), while the same letters mean no significant differences (*p* > 0.05).

**Figure 7 ijms-23-07551-f007:**
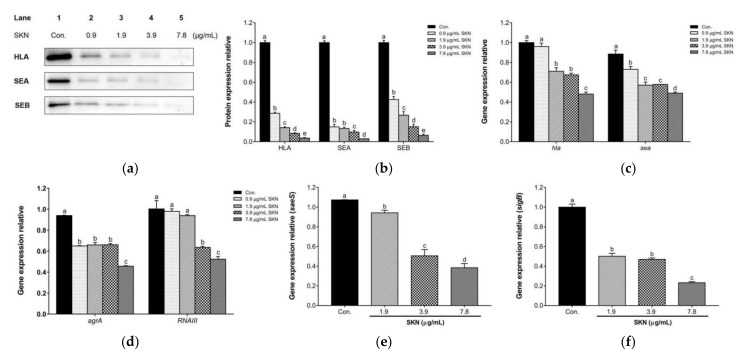
(**a**) The effect of SKN on the protein expression of α-hemolysin (HLA), staphylococcal enterotoxin A (SEA) and staphylococcal enterotoxin B (SEB) in clinical MRSA strain DPS-1, as analyzed by Western blot. (**b**) The intensities of Western blot protein bands were quantified using ImageJ software. (**c**) The effect of SKN on the mRNA expression of genes encoding HLA (hla) and SEA (sea), as analyzed by qRT-PCR. The effect of SKN on the mRNA expression of regulatory genes agrA (**d**), RNAIII (**d**), saeS (**e**), and sigB (**f**) in clinical MRSA strain DPS-1. The data are means ± SD of triplicate determinations. Different letters on the bars indicate significant statistical differences between treatments (*p* < 0.05), while the same letters mean no significant differences (*p* > 0.05).

**Figure 8 ijms-23-07551-f008:**
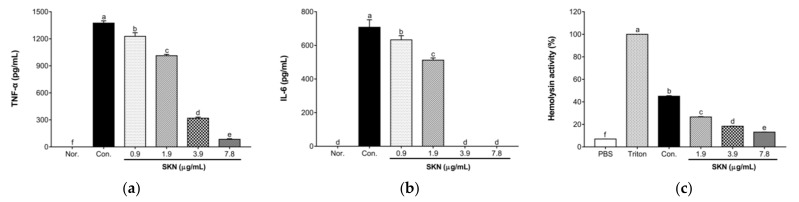
ELISA analysis of tumor necrosis factor α (TNF-α) (**a**) and interleukin-6 (IL-6) (**b**) released by RAW 264.7 cells stimulated with the supernatants of MRSA DPS-1 grown in the presence of SKN. (**c**) The effect of SKN on the hemolytic activity of clinical MRSA strain DPS-1. The data are means ± SD of triplicate determinations. Different letters on the bars indicate significant statistical differences between treatments (*p* < 0.05), while the same letters mean no significant differences (*p* > 0.05).

**Figure 9 ijms-23-07551-f009:**
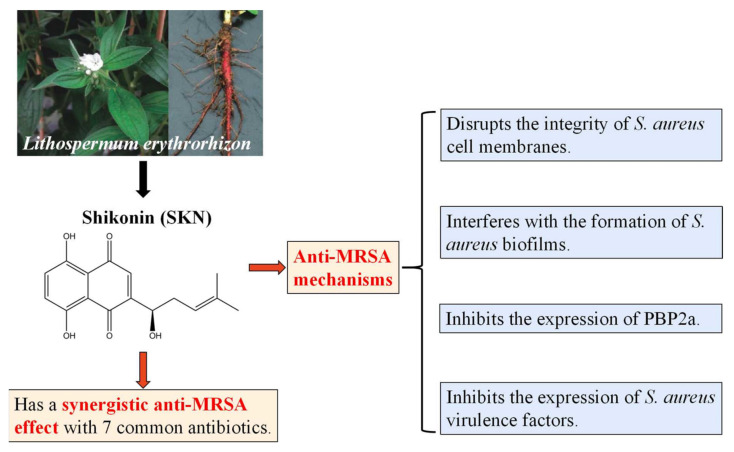
Schematic diagram of the antibacterial activity of shikonin (SKN) found in this study.

**Table 1 ijms-23-07551-t001:** Minimum inhibitory concentration (MIC) and fractional inhibitory concentration index (FICI) of shikonin (SKN) and seven antibiotics against MRSA ATCC 33591, CCARM 3090 and DPS-1.

Agents	ATCC 33591				CCARM 3090				DPS-1			
	MIC (µg/mL)		Fold	FICI	MIC (µg/mL)		Fold	FICI	MIC (µg/mL)		Fold	FICI
	Alone	Comb.			Alone	Comb.			Alone	Comb.		
SKN	15.6	7.8	2	1	125	7.8	16	0.56	15.6	3.9	4	0.5
OXA	31.25	15.6	2		250	125	2		31.25	7.8	4	
SKN	15.6	3.9	4	0.5	125	7.8	16	0.19	15.6	1.9	8	0.37
GEN	7.8	1.9	4		250	31.25	8		125	31.25	4	
SKN	15.6	1.9	8	0.37	125	7.8	16	0.31	15.6	1.9	8	0.37
AMI	62.5	15.6	4		7.8	1.9	4		62.5	15.6	4	
SKN	15.6	0.9	16	0.31	125	31.25	4	0.5	15.6	0.9	16	0.56
AMO	250	62.5	4		62.5	15.6	4		62.5	31.25	2	
SKN	15.6	0.9	16	0.56	125	7.8	16	0.31	15.6	1.9	8	0.37
CEFO	31.25	15.6	2		62.5	15.6	4		31.25	7.8	4	
SKN	15.6	3.9	4	0.75	125	7.8	16	0.31	15.6	0.9	16	0.31
CEFT	125	62.5	2		1,000	250	4		250	62.5	4	
SKN	15.6	3.9	4	0.75	125	7.8	16	0.56	15.6	0.9	16	0.31
CHL	125	62.5	2		7.8	3.9	2		250	62.5	4	

Values are means of three independent experiments. Index interpretation: <0.5, synergy; 0.5–0.75, partial synergy; 0.75–1, additive effect; 1–4, no effect. Comb, combination; ATCC, American Type Culture Collection; CCARM, Culture Collection of Antimicrobial Resistant Microbes; DPS, Department of Plastic Surgery; OXA, oxacillin; GEN, gentamicin; AMI, amikacin; AMO, amoxicillin; CEFO, cefoxitin; CEFT, ceftazidime; CHL, chloramphenicol.

**Table 2 ijms-23-07551-t002:** Sequences of oligonucleotide primers designed for qRT-PCR.

Gene	Forward Primer (5′–3′)	Reverse Primer (5′–3′)
*icaA*	CAATACTATTTCGGGTGTCTTCACTCT	CAAGAAACTGCAATATCTTCGGTAATCAT
*fnbA*	AACTGCACAACCAGCAAATG	TTGAGGTTGTGTCGTTTCCTT
*mecA*	GCAATCGCTAAAGAACTAAG	AATGGGACCAACATAACCTA
*mecR1*	ACACGACTTCTTCGGTTAG	GTACAATTTGGGATTTCACT
*hla*	TTGGTGCAAATGTTTC	TCACTTTCCAGCCTACT
*sea*	ATGGTGCTTATTATGGTTATC	CGTTTCCAAAGGTACTGTATT
*agrA*	TGATAATCCTTATGAGGTGCTT	CACTGTGACTCGTAACGAAAA
*RNAIII*	GCACTGAGTCCAAGGAAACTAAC	AAGCCATCCCAACTTAATAACC
*saeS*	TGTATTTAAAGTGATAATATGAGTC	CTTAGCCCATGATTTAAAAACACC
*sigB*	AAGTGATTCGTAAGGACGTCT	TCGATAACTATAACCAAAGCCT
*16S*	ACTCCTACGGGAGGCAGCAG	ATTACCGCGGCTGCTGG

## Data Availability

Not applicable.

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
