# Peer review of "Synergistic Antibacterial Activity with Conventional Antibiotics and Mechanism of Action of Shikonin against Methicillin-Resistant *Staphylococcus aureus"

_ijms, 2022, doi:10.3390/ijms23147551_

Round 1

Reviewer 1 Report

The manuscript presented for review is well-written and structured report of the shokonin as a potential treatment for MRSA infection. Methicillin resistant Staphylococcus aureus strains pose a serious treatment problems among hospitalized patients. Moreover, bacteria resistant to glycopeptides antibiotics, which are the drugs of the last resort against MRSA, have been identified. An increased resistance to macrolide, lincosamide and streptogramin B antibiotics among staphylococci is also observed, what is a consequence of their extensive use in infections caused by Gram-positive bacteria. The clinical environment modifies the disease progression, which results in prolonged hospitalization (up to 5-10 days), an increase in the mortality rate and an increase in treatment costs from 30% to even 100%.

The methodology used by the authors is appropriate. The results are presented in a very precise way. Overall, this is a clear and well organized manuscript and it is necessary to do only minor revision in the text before acceptation. 

I suggest authors consider using the full name in the title instead of the "MRSA" abbreviation

Line 25: anti-bacterial à antibacterial

Line 53: (L.erythrorhizon) à (L. erythrorhizon)

Line 57: L.erythrorhizon à L. erythrorhizon

Line 82: no spaces before the parentheses - gentamicin(GEN)

Line 83-84: no spaces before the parentheses: amoxicillin(AMO), cefoxitin(CEFO), ceftazidime(CEFT), and chloramphenicol(CHL)

Line 83: ( AMI) à there is extra space before AMI

Line 263: almost all à all

Author Response

Please refer to the attached file'

Reviewer 2 Report

In this manuscript the authors explore the use of shikonin (SKN) as an alternative to traditional antibiotics. They showed that SKN can be use in synergy with clinical antibiotics and can improve their activity. Also, the authors showed that SKN can reduce the expression of important MRSA virulence factors such as α-hemolysin. 

This is an interesting study that contains enough novelty for publication in Int. J. Mol. Sci.

However there few minor issues that the authors should improve before publication:

i) The title is too long;

ii) Introduction section, " S. aureus is second only to pneumococcus among the common causes of pneumonia in African children"; S aureus in fact is one of major causes of bacterial infections found in hospitals worldwide. Please improve this sentence.

iii) Please add a figure summarizing the major finds depicted in this study.

Author Response

Please refer to the attached file'
